∂ | **Open Peer Review** | Environmental Microbiology | Research Article

# Organic matter lability modifies the vertical structure of methane-related microbial communities in lake sediments

Antti J. Rissanen,[1,2] Tom Jilbert,[3] Asko Simojoki,[4] Rahul Mangayil,[1] Sanni L. Aalto,[5,6] Ramita Khanongnuch,[1] Sari Peura,[7] Helena Jäntti[5]

**ABSTRACT** Eutrophication increases the input of labile, algae-derived, organic matter (OM) into lake sediments. This potentially increases methane ($CH_4$) emissions from sediment to water through increased methane production rates and decreased methane oxidation efficiency in sediments. However, the effect of OM lability on the structure of methane oxidizing (methanotrophic) and methane producing (methanogenic) microbial communities in lake sediments is still understudied. We studied the vertical profiles of the sediment and porewater geochemistry and the microbial communities (16S rRNA gene amplicon sequencing) at five profundal stations of an oligo-mesotrophic, boreal lake (Lake Pääjärvi, Finland), varying in surface sediment OM sources (assessed via sediment C:N ratio). Porewater profiles of methane, dissolved inorganic carbon (DIC), acetate, iron, and sulfur suggested that sites with more autochthonous OM showed higher overall OM lability, which increased remineralization rates, leading to increased electron acceptor (EA) consumption and methane emissions from sediment to water. When OM lability increased, the abundance of anaerobic nitrite-reducing methanotrophs (*Candidatus* Methylomirabilis) relative to aerobic methanotrophs (*Methylococcales*) in the methane oxidation layer of sediment surface decreased, suggesting that *Methylococcales* were more competitive than *Ca*. Methylomirabilis under decreasing redox conditions and increasing methane availability due to their more diverse metabolism (fermentation and anaerobic respiration) and lower affinity for methane. Furthermore, when OM lability increased, the abundance of methanotrophic community in the sediment surface layer, especially *Ca*. Methylomirabilis, relative to the methanogenic community decreased. We conclude that increasing input of labile OM, subsequently affecting the redox zonation of sediments, significantly modifies the methane producing and consuming microbial community of lake sediments.

**IMPORTANCE** Lakes are important natural emitters of the greenhouse gas methane ($CH_4$). It has been shown that eutrophication, via increasing the input of labile organic matter (OM) into lake sediments and subsequently affecting the redox conditions, increases methane emissions from lake sediments through increased sediment methane production rates and decreased methane oxidation efficiency. However, the effect of organic matter lability on the structure of the methane-related microbial communities of lake sediments is not known. In this study, we show that, besides the activity, also the structure of lake sediment methane producing and consuming microbial community is significantly affected by changes in the sediment organic matter lability.

**KEYWORDS** greenhouse gas, freshwater, methanotroph, methanogen, 16S rRNA gene, eutrophication

Address correspondence to Antti J. Rissanen, antti.rissanen@tuni.fi.

The authors declare no conflict of interest.

See the funding table on p. 12.

The concentration of atmospheric methane ($CH_4$), a critical greenhouse gas, has increased substantially since industrialization, with current total emissions of 550–

600 Tg/year (top-down estimates) of which approximately 40% stem from natural sources (1). Lakes are important natural emitters of $CH_4$. The numerous lakes and ponds of the northern boreal zone, with annual emissions of ~16.5 Tg, are especially important contributors to the global $CH_4$ budget (2, 3). Thus, knowledge of the factors affecting $CH_4$ emissions from northern lakes is essential for accurate estimates and modeling of the global $CH_4$ budget and its changes due to global change (e.g., eutrophication and climate warming).

The $CH_4$ emissions from lake sediments are controlled by the balance between methanogenesis and $CH_4$ oxidation. Methanogenesis is the final step in anoxic organic matter (OM) degradation by methanogenic archaea that produce $CH_4$ from acetate or from oxidation of $H_2$ using $CO_2$ as an electron acceptor (EA) (4). Some methanogens can also produce $CH_4$ from other methyl compounds (e.g., methanol) (4, 5). In oxic surface sediments, $CH_4$ is consumed through aerobic $CH_4$ oxidation by methanotrophic bacteria (MOB) using $O_2$ as an EA (6). In anoxic conditions, $CH_4$ is consumed through anaerobic methane oxidation (AOM) by anaerobic methanotrophic archaea (ANME) utilizing various inorganic or organic compounds (7–9) or by bacteria within genus *Candidatus* Methylomirabilis utilizing $NO_2^-$ as EAs (10). Furthermore, MOB belonging to *Methylococcales* have been shown to be capable of metabolizing $CH_4$ in hypoxic and anoxic conditions via fermentation and anaerobic respiration of various EAs, including $NO_3^-$, $NO_2^-$, $Fe^{3+}$, and organic EAs (11–19).

Eutrophication was recently shown to increase $CH_4$ emissions from lake sediments through lowered $CH_4$ oxidation efficiency at increasing sediment methanogenesis rate (20). However, whether this phenomenon is due to change in the activity of methanogens and methanotrophs or also due to change in their community structure and abundance is not known. In support of the latter hypothesis, the results from the recent studies comparing lakes with different trophic states suggest that the density of sediment methanogens is higher while that of methanotrophs is lower, in eutrophic lakes that contain more labile algae-based sediment OM and where $CH_4$ fluxes from the sediment are higher than in oligotrophic and mesotrophic lakes (21–23). However, no previous study has simultaneously studied the change in both methanotrophic and methanogenic communities within a gradient of changing OM lability. The close spatial proximity of MOB and AOM-driving *Ca.* Methylomirabilis bacteria in lake sediments (24) also suggests that competition for $CH_4$ exists between these groups. Indeed, a recent modeling study suggested that an increase in OM quantity decreases the abundance of *Ca.* Methylomirabilis while increasing the abundance of MOB (25). Furthermore, in the comparison [using the 16S rRNA gene dataset by Han et al. (21)] of sediment methanotroph communities between lakes with varying trophic status by van Grinsven et al. (22), *Ca.* Methylomirabilis sp. was as abundant as MOB in the sediments of an oligotrophic lake but had negligible abundance in the sediments of meso- and eutrophic lakes, where MOB dominated the methanotrophic community. These differences suggest that MOB are more competitive in conditions of higher OM lability, when increased OM mineralization decreases redox potential and increases $CH_4$ availability. This is possibly due to MOB, especially *Methylococcales*, having a lower affinity for $CH_4$ and more diverse metabolism in hypoxic and anoxic conditions (incl. capability for fermentation and anaerobic respiration of various EAs as explained above) than *Ca.* Methylomirabilis (12, 15, 17, 18, 26–28). Hence, it could be expected that changes in the OM quality of lake sediments affect differently the abundances of MOB and *Ca.* Methylomirabilis sp. bacteria.

In this study, we investigated how spatial variability in OM lability in a single lake affects the community structure of $CH_4$ producing and consuming sedimentary microbes. Using 16S rRNA gene sequencing, we studied the variation in the methanogenic and methanotrophic community in the uppermost sediment layers at five sites with naturally varying OM quality of surface sediment within an oligo-mesotrophic boreal Lake Pääjärvi, Southern Finland. To determine OM lability, we used bulk sediment C:N ratios, which have been shown to broadly reflect relative contributions

of (autochthonous) phytoplankton and (allochthonous) terrestrial sources to bulk sedimentary OM (29, 30). Higher C:N ratios indicate a greater proportion of terrestrial OM, which is considered less labile (less available for microbial degradation) due to a combination of primary chemical composition, degradation during transport and protection by aggregation with mineral material [(31) and references therein], whereas lower C:N indicates more labile, phytoplankton-derived material [(32) and references therein]. We hypothesized that under increasing OM lability, (i) the relative contribution of methanotrophs within the $CH_4$ cycling community decreases while that of methanogens increases and (ii) the relative contribution of *Ca*. Methylomirabilis sp. bacteria within the $CH_4$ oxidizing community decreases while that of MOB increases.

## MATERIALS AND METHODS

### Study lake

Lake Pääjärvi is a $NO_3^-$-rich, oligo-mesotrophic lake in Southern Finland (61.04N, 25.08E; A = 13.4 km$^2$, max. depth = 87 m, mean residence time = 3.3 yr). The water column circulates twice per year and is always well oxygenated (33). Field measurements of dissolved oxygen during this study (determined with a handheld YSI ODO probe) confirmed the presence of oxygen throughout the water column at all sediment sampling locations (Fig. S1). The hypolimnetic $NO_3^-$ concentration, measured 2–5 cm above the sediment surface, is typically 46–75 µmol L$^{-1}$ (34, 35). The large catchment area of Lake Pääjärvi (244 km$^2$) is dominated by forests and agriculture. The nutrient concentrations of the water have increased since the 1970s (36).

### Porewater and sediment sampling

Vertical profiles of porewater and sediment samples were collected from five stations in Lake Pääjärvi on 9 August 2017 using a handheld HTH/Kajak corer with plexiglass tubes (Table 1). This study uses data from the top-most 10 cm layer (i.e., 0–10 cm from the sediment-water interface surface). The study stations followed a water depth gradient, Station 1 being the shallowest and Station 5 the deepest (Table 1). The core tubes were pre-drilled with two vertical series of 4 mm holes (each at 2 cm resolution), and then taped, in preparation for porewater sampling with Rhizons (Rhizosphere research products, Wageningen, Netherlands). Rhizon sampling automatically filters the porewaters at 0.15 µm into attached 10 mL syringes under vacuum. One vertical series of samples was taken for analysis of dissolved $CH_4$ and dissolved inorganic carbon (DIC). The syringes were pre-filled with 1 mL of 0.1M $HNO_3$ to immediately convert all DIC to $CO_2$. A known volume of $N_2$ gas headspace was injected into the syringes after sampling, and the samples were shaken to equilibrate the dissolved gases with the headspace. The subsamples of the headspace were then extracted into 3 mL Exetainers (Labco Limited, Lampeter, UK) and stored at room temperature (RT) until analysis [for full details see (37)]. The second vertical series was taken for S and Fe (by inductively coupled plasma atomic emission spectroscopy, ICP-OES) and short chain organic acid analysis. Short chain organic acid subsamples were stored frozen at −20°C until analysis. The subsamples for ICP-OES were acidified with 1 M $HNO_3$ and stored at RT. After porewater sampling, the sediment cores were sliced into plastic bags at a resolution of 1 cm. The subsamples of 400−500 µL wet sediment were collected from each slice and stored frozen at −20°C for DNA-based molecular microbiological analyses. The remaining wet sediment samples were stored frozen at −20°C under $N_2$ until further processing.

### Porewater and sediment bulk geochemical analysis

The S and Fe concentrations in the porewater samples were determined by ICP-OES (Thermo iCAP 6000, Thermo Fisher Scientific, Waltham, MA, USA). In this system, S is expected to be dominated by sulfate ($SO_4^{2-}$) and Fe by $Fe^{2+}$, although in each case, other forms are possible. The porewater $CH_4$ and DIC concentrations were determined

**TABLE 1** C:N ratios of the surface sediment [0–1 cm layer and 0–2 cm layer (i.e., average of 0–1 cm and 0–2 cm layers)], sediment porewater acetate and dissolved inorganic carbon (DIC) concentrations, and Shannon diversity index of prokaryotic diversity in the sediment (average +/−SD within the 0–10 cm layer) as well as the estimated $CH_4$ emissions from sediment to water (based on porewater $CH_4$ profiles) at the study stations[b]

| Station | Depth (m) | C:N (0–1) | C:N (0–2) | Acetate (µmol L⁻¹) | SD | DIC (µmol L⁻¹) | SD | CH₄ flux (µmol m⁻² d⁻¹) | Shannon | SD |
|---------|-----------|-----------|-----------|--------------------|----|----|-----|------------------------|---------|-----|
| 3 | 52 | 14.74 | 14.73 | 9.3 | 8.5 | 553.4 | 269.3 | 280.6 | 6.71 | 0.16 |
| 2 | 22 | 13.40 | 13.78 | 5.3 | 7.3 | 589.7 | 322.6 | 99.5 | 6.80 | 0.21 |
| 4 | 60 | 12.90 | 13.38 | 15.4 | 1.3 | 675.8 | 140.0 | 1,148.2 | 6.93 | 0.11 |
| 1 | 14 | 12.78 | 13.25 | 13.7 | 2.5 | 900.8 | 93.0 | 1,262.6[a] | 7.01 | 0.10 |
| 5 | 80 | 12.54 | 12.50 | 14.4 | 0.8 | 965.4 | 144.8 | 4,324.8[a] | 6.99 | 0.13 |

[a]CH₄ flux estimates for stations 1 and 5 should be considered as maximum estimates because the sampling resolution was lower at the top of the core (due to geometry of the core in the tube) and, therefore, the gradient at the sediment-water-interface may be less steep in reality (see Fig. 1).
[b]Study stations are organized in the order of increasing organic matter lability based on decreasing surface sediment C:N ratios. See full vertical profiles (for the 0–10 cm layer) of C:N ratio, DIC, acetate, and Shannon diversity index in Fig. S2A through D, respectively.

by gas chromatography (GC) as described in Jilbert et al. (37). Briefly, the sample vials (Exetainers) were pressurized with helium (He) to 2.0 bar before loading into the GC (Agilent technologies 7890B GC system, Agilent Technologies, Santa Clara, Ca, USA). $CH_4$ was determined by a flame-ionization detector (FID) and $CO_2$ by a thermal conductivity detector (TCD). The instrument simultaneously measures $N_2$ and $O_2$ + Ar (in TCD), from which a 100% sum can be calculated for the estimation of $CH_4$ and $CO_2$ concentrations in the original sample, in ppm by volume. For a full description of the calculations, see Jilbert et al. (37). Diffusive fluxes of methane across the sediment-water-interface were estimated using Fick's Law:

$$F = \frac{\varphi \cdot D}{\theta^2} \frac{\partial C}{\partial z}$$

where $F$ = flux in µmol m⁻² d⁻¹, $D$ = diffusion coefficient of $CH_4$ in freshwater, based on a value of $1.67 \times 10^{-9}$ m² s⁻¹ at 25°C and adjusted downward according to bottom water temperature, using Eq. 4.57 in Boudreau (38); $\varphi$ = volume fraction of total porosity and $\theta$ = tortuosity, as related by $\theta^2 = 1 - \ln(\varphi^2)$; and $\partial C/\partial z$ is the partial differential gradient estimated from the finite difference gradient $\Delta C/\Delta z$, the concentration gradient of methane between the uppermost porewater sample and the overlying water in the sediment core tube.

Sediment samples were freeze-dried, grounded in an agate mortar, and weighed into tin cups for carbon (C) and nitrogen (N) content determinations, which were determined using an elemental analyzer (LECO TruSpec Micro, LECO Corp., St. Joseph, MI, USA). In accordance with extensive previous studies on Finnish lake sediments (39, 40), acidification was not applied prior to the determinations. High levels of organic acidity from Finnish river catchments (41) maintain low annual mean pH values in most lakes and, therefore, there is a negligible occurrence of carbonates in lake sediments. Hence, our total C data is considered equivalent to organic C ($C_{org}$), and total N is considered equivalent to organic N ($N_{org}$).

Porewater short chain organic acids were analyzed using high performance liquid chromatography (HPLC) equipped with Shodex SUGAR column (300 mm × 8 mm, Phenomenex, Torrance, CA, USA), autosampler (SIL-20AC HT, Shimadzu, Kioto, Japan), refractive index detector (RID-10A, Shimadzu), and 0.01 M $H_2SO_4$ as the mobile phase. The HPLC samples were prepared as described in Salmela et al. (42). The identification and quantification of the liquid metabolites were conducted using external standards.

## Molecular microbiological analyses

DNA was extracted from the frozen sediment samples using DNeasy PowerSoil Kit (Qiagen, Hilden, Germany). DNA concentration was measured using a Qubit 2.0 Fluorometer and a dsDNA HS Assay Kit (Thermo Fisher Scientific, Waltham, MA, USA).

PCR and 16S rRNA gene amplicon sequencing was performed commercially by the Foundation for the Promotion of Health and Biomedical Research of Valencia Region

(FISABIO, Valencia, Spain). In the PCR reactions, the V4 region of the bacterial and archaeal 16S rRNA genes was simultaneously targeted using primer pair 515FB (5′-GTGYCAGCMGCCGCGGTAA-3′)/806FB (5′-GGACTACNVGGGTWTCTAAT-3′) (43, 44). PCR, library preparation, and paired-end sequencing (Illumina MiSeq, Illumina, San Diego, CA, USA) were performed as previously described (45), except that, in PCR reactions, approximately 15 ng of DNA were used.

The quality assessment of the raw sequence reads, merging of paired-end reads, alignment, chimera removal, preclustering, taxonomic classification (using Silva database 132), and removal of chloroplast, mitochondria, and eukaryote sequences, and division of 16S rRNA gene sequences into operational taxonomic units (OTUs) at 97% similarity level was conducted as described in Rissanen et al. (11) (the detailed description is available also in Supplementary Methods). Singleton OTUs (OTUs with only one sequence) were removed, and the data were then normalized by subsampling to the same size, 77,340 sequences. One sample, representing the layer 8–9 cm depth at Station 3, was discarded from the analyses since it had only ~10,000 joined sequence reads. Good coverage was 0.95–0.97 in each library confirming that sequence variation was adequately covered. Prokaryotic diversity was assessed via calculation of the Shannon diversity index for each library. To test the hypotheses, this study focused specifically on the relative abundance (% of prokaryotic 16S rRNA genes) of known aerobic (46) and anaerobic methanotrophic bacteria (10), anaerobic methanotrophic archaea (47) as well as methanogenic archaea (5, 48).

## Statistical analyses

The relationship between microbial variables and the OM quality of sediment was analyzed using the Spearman's rank correlation test. Data showing significant correlations ($P < 0.05$) were further studied using linear and non-linear regression analyses. The microbial variables included the Shannon diversity index and the relative abundances (percentage of 16S rRNA genes) of methanotrophs and methanogens (both total and different taxonomic groups), while C:N ratio of the surface sediment (0–1 and 0–2 cm layer) was used as a proxy for the OM quality of sedimenting OM (i.e., decreasing C:N ratio indicates increasing lability of sedimenting OM). Furthermore, sediment porewater DIC concentration was used as a further indicator of sediment OM lability in the correlation analyses (i.e., increasing DIC indicates increasing OM mineralization due to increased OM lability). We acknowledge that the relative abundances of microbes do not predict their absolute abundances. However, ratios of the relative abundances of different organisms to each other are robust against variations in their absolute abundances. Therefore, besides relative abundances, we also analyzed abundance ratios of microbes, such as the ratio of *Ca.* Methylomirabilis/*Methylococcales*, Methanotrophs/Methanogens, and *Ca.* Methylomirabilis/Methanogens. Correlation analyses were done using IBM SPSS Statistics 26 (IBM SPSS Statistics for Windows, Version 26.0, IBM Corp., Armonk, NY, USA), while the regression analyses were done using Minitab software (Minitab Statistical Software for Windows, Version 16.2.0.0, Minitab Inc., PA, USA). To determine the best-fitting models to the experimental data, along with their 95% confidence intervals (CIs), the parameter values of models were adjusted so as to minimize the squared deviation between the data values and the fitted values (*S*) and via checking the normality of residuals (*P*-value > 0.05).

## RESULTS AND DISCUSSION

### Sediment OM quality, vertical porewater profiles, and prokaryotic diversity

Stations were ordered according to increasing OM lability as determined by decreasing C:N ratio in surface sediment as follows: Station 3, Station 2, Station 4, Station 1, and Station 5 (Table 1; Fig. S2A). This ordering did not follow the depth gradient, and therefore likely represents the heterogeneity of sedimentation of autochthonous and allochthonous OM within the lake. Surface sediment values were used here due to

assumed minimum overprinting of sediment diagenetic processes on the C:N ratio; hence, the values should represent the C:N ratio (lability) of the OM sedimenting to the lake bottom at the stations. Using C:N ratio of either the 0–1 cm layer or the average of 0–1 cm and 0–2 cm layers had no impact on the order of stations in terms of OM lability (Table 1; Fig. S2A). Hence, the possible mixing of surface sediment by bioturbation did not affect the major pattern in the OM lability between stations. The range of observed molar C:N ratios is quite narrow (approx 12.5–14.7 for the 0–1 cm interval) indicating an overall dominance of allochthonous terrestrial OM according to the end-member values of Goñi et al. (29), but small differences in the contribution of autochthonous OM appear to strongly influence the overall OM lability. Evidence for an OM lability gradient was shown in porewater DIC and acetate data. Average concentration of DIC within the 0–10 cm layer increased when there was a decrease in surface sediment C:N (Table 1, Fig. S2B) ($\rho = -1.00$, $P < 0.0001$, for both C:N ratio of 0–1 cm layer and average of C:N ratio of 0–1 and 0–2 cm layers). Although not significantly correlated with surface sediment C:N ratio ($\rho = -0.6$, $P = 0.285$, for both C:N ratio of 0–1 cm layer and average of C:N ratio of 0–1 and 0–2 cm layers), acetate was also higher in stations with lower (i.e., Stations 4, 1, and 5) than with higher surface sediment C:N (Stations 3 and 2) (Table 1; Fig. S2C). Lactate was also detected but only within the sediments of Station 1, which was among the stations with the lowest surface sediment C:N ratio (Fig. S2C). These results suggest that OM remineralization rates (producing DIC, acetate, and lactate) were higher in stations with lower surface sediment C:N, which reflects their higher sediment OM lability.

According to visual analysis of the porewater data, the differences in the vertical profiles of $CH_4$, S (assumed to be mostly $SO_4^{2-}$) and Fe (assumed to be mostly $Fe^{2+}$ from reduction of solid-phase Fe oxides) between the stations followed broadly the differences in OM lability (Fig. 1). $CH_4$ concentrations increased from the sediment surface downward with depth at each station (Fig. 1A through E). However, compared to other stations, they remained at very low levels at surface 0–4 cm layer at Stations 3 and 2 with the lowest sediment OM lability, while the highest surface sediment $CH_4$ concentrations were observed at Station 5 with the highest OM lability (Fig. 1A through E). This pattern generally agrees with the comparison of sediment $CH_4$ profiles between eutrophic (labile sediment OM) and oligotrophic (less labile OM) lakes by van Grinsven et al. (22). This suggests that active zones of methanogenesis extend closer to the sediment surface, and $CH_4$ oxidation takes place in a thinner surface layer in stations with higher sediment OM lability than in those with lower OM lability (Fig. 1A through E). In addition, S (i.e., $SO_4^{2-}$, see above) and Fe ($Fe^{2+}$, see above) profiles suggested that the zone of reduction of $SO_4^{2-}$ and Fe oxides were located deeper from the sediment surface in the stations with less labile OM, as seen in S accumulation zone extending deeper and Fe accumulation zone starting deeper from the sediment surface in stations with less labile OM (Fig. 1F through O). As the water column was well oxygenated at each station (Fig. S1), the differences in the porewater profiles of $CH_4$, S, and Fe between stations are not explained by differences in oxygen availability in the water overlying the sediment but are very likely driven by the differences in the sediment OM lability. Altogether, the porewater profiles suggested a higher rate of OM processing, higher consumption of $O_2$ and alternative EAs, lower redox potential, and subsequently higher rate of methanogenesis in the stations with higher sediment OM lability (Fig. 1). This agrees with previous results on the comparison of lakes with different trophic status (22, 23). In accordance, the modeled diffusive methane emissions from sediment to water were generally higher in the stations with higher sediment OM lability (Table 1) ($\rho = -0.90$, $P < 0.05$, for both C:N ratio of 0–1 cm layer and average of C:N ratio of 0–1 and 0–2 cm layers), which agrees with the results by van Grinsven et al. (22). The magnitudes of estimated methane emissions from sediment to water, 99.5–4324.8 $\mu mol\ m^{-2}\ d^{-1}$ (Table 1), agree well with the range of previously measured data for boreal lakes, i.e., 300–6562.5 $\mu mol\ m^{-2}\ d^{-1}$ (49).

Prokaryotic diversity, assessed via analysis of Shannon diversity index, increased when sediment OM lability increased (i.e., when C:N ratio decreased) (Table 1; Fig. S2D) ($\rho = -0.90$, $P < 0.05$, for both C:N ratio of 0–1 cm layer and average of C:N ratio of 0–1 and

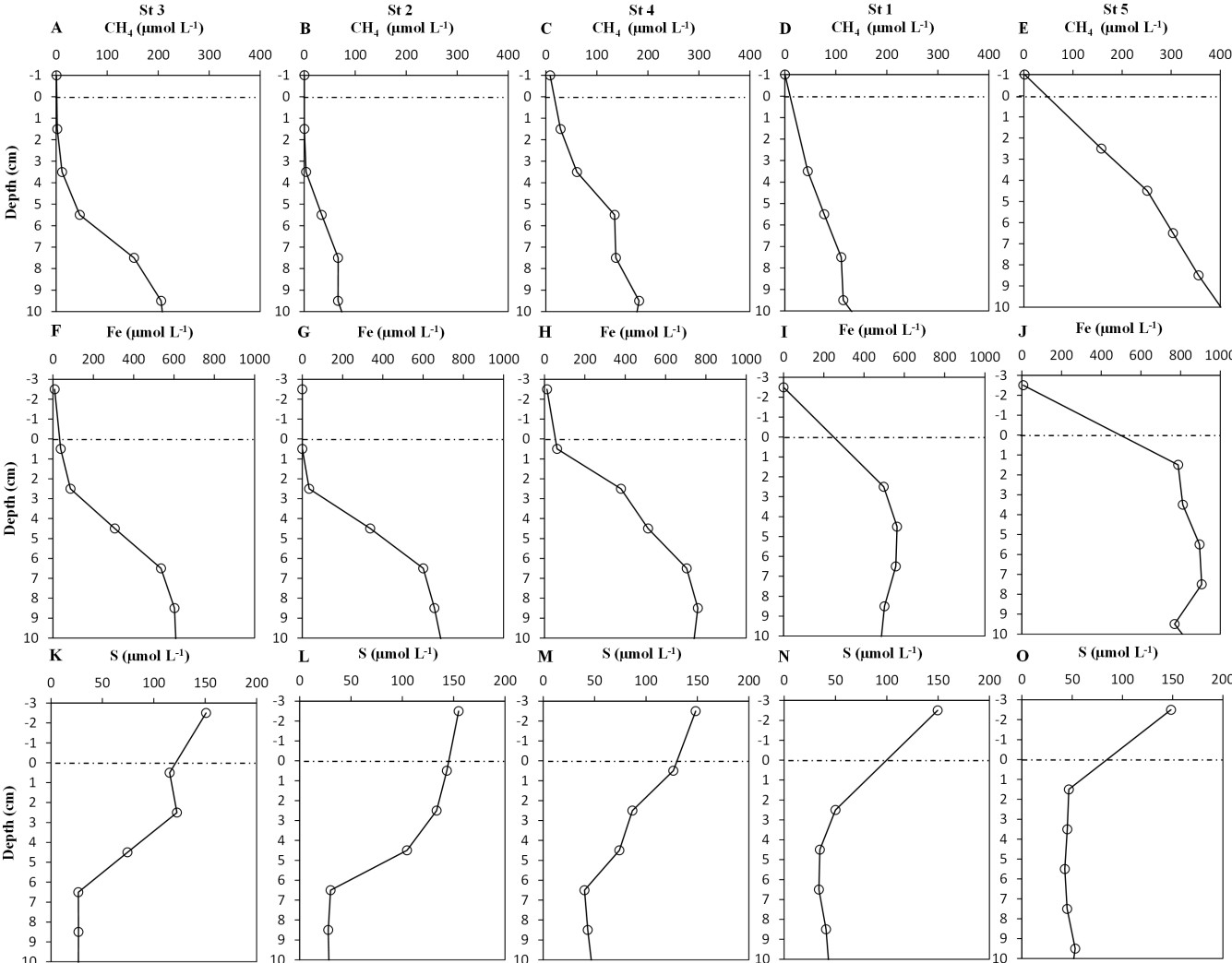

**FIG 1** Concentrations of $CH_4$ (A-E), Fe (F-J), and S (K-O) in the porewater at different depths of sediments (incl. water overlying the sediment) at the five study stations shown in the order of increasing OM lability (from left to right; St3, St2, St4, St1, and St5) based on surface sediment C:N ratio (see Table 1). Depth 0 cm (dashed line) indicates the sediment-water interface.

0–2 cm layers). This result agrees with those from soil ecosystems showing negative correlation between bacterial diversity and soil C:N ratio (50). This can be explained by high resource quality (i.e., labile OM with low C:N ratio) leading to a greater variety of resources for bacterial communities, which enhances their diversity by promoting greater niche differentiation (50, 51). Hence, variation in the prokaryotic diversity further highlights the differences in OM lability between stations.

## Methanotrophic community

Aerobic MOB in the order *Methylococcales* as well as anaerobic methanotrophs in genera *Ca.* Methylomirabilis sp. (within bacterial phylum *NC10*) and *Ca.* Methanoperedens sp. (also known as ANME 2D archaea) were the most abundant groups of methanotrophs present in the studied sediments (Fig. 2A through E). This agrees with a previous study on boreal lake sediments (52). Aerobic MOB in the family *Methylacidiphilaceae* (i.e., Verrucomicrobial methanotrophs) were rare, while alphaproteobacterial MOB were not detected (Fig. 2A through E). The relative abundance of both *Methylococcales* and *Ca.* Methylomirabilis sp. generally peaked in the surface 0–3 cm layer at each station, while *Ca.* Methylomirabilis was occasionally observed also at deeper depths (Fig. 2A through

E). In contrast to *Methylococcales* and *Ca*. Methylomirabilis sp., *Ca*. Methanoperedens sp. archaea had peaks in its relative abundance only at deeper layers, clearly below the sediment surface (i.e., below 5 cm depth) (Fig. 2A through E). In the deep layers, *Ca*. Methanoperedens sp. archaea can potentially use a wide variety of EAs in AOM, i.e., $SO_4^{2-}$, $Fe^{3+}$ minerals, and organic compounds (8, 47, 53–56), but as the scope of this study is focused on *Ca*. Methylomirabilis sp. and MOB, the putative role of *Ca*. Methanoperedens sp. archaea in the biogeochemical processes of the study lake is not considered further.

The $CH_4$ profiles in porewater show clearly depleted $CH_4$ concentrations near the sediment surface (Fig. 1A through E), suggesting that abundant *Methylococcales* and *Ca*. Methylomirabilis sp. act as a crucial filter in the top 0–3 cm sediment layer, reducing $CH_4$ fluxes from sediment to the overlying water column (Fig. 2A through E). The overlap in the vertical distribution of *Methylococcales* and *Ca*. Methylomirabilis sp. agrees with previous results from lake sediments and suggests competition between these groups (22, 24), as further indicated by a recent modeling study (25). Therefore, we tested the hypothesis that the relative contribution of *Ca*. Methylomirabilis sp. and MOB within the $CH_4$ oxidizing community is sensitive to OM lability. In this analysis, we specifically focused on the top 0–3 cm sediment layer, as based on depleted $CH_4$ concentrations and high methanotroph relative abundances at that layer, it is considered to be the key $CH_4$ filter layer in reducing $CH_4$ emissions from sediment to water (Fig. 1A through E and 2A through E). In partial support for the hypothesis, we found that within that layer, the relative abundance of *Ca*. Methylomirabilis decreased, when OM lability increased, i.e., *Ca*. Methylomirabilis correlated positively with sediment C:N ratio (Table 2; Fig. 3A), while the relative abundance of *Methylococcales* was not correlated with the C:N ratio (Table 2). Consequently, the ratio of *Ca*. Methylomirabilis to *Methylococcales* decreased alongside when OM lability increased (C:N decreased) (Table 2; Fig. 3B). As it is challenging to quantify the relative abundance of taxonomic groups in a layer (i.e., 0–3 cm) consisting of multiple sublayers (i.e., 0–1 cm, 0–2 cm, and 0–3 cm), we increased the robustness in our results by carrying out these analyses both with the average relative abundance and the maximum relative abundance of *Methylococcales* and *Ca*. Methylomirabilis within the 0–3 cm sediment layer, with both choices giving similar results for the change in *Ca*.

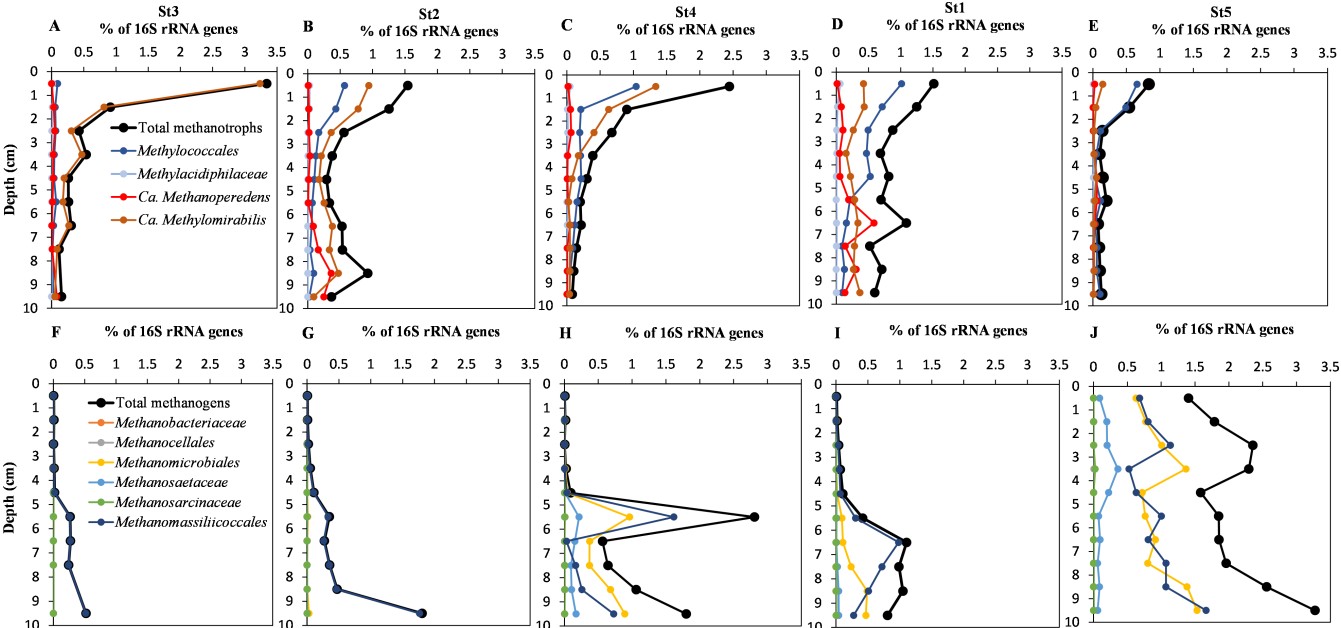

**FIG 2** Relative abundance (percentage of prokaryotic 16S rRNA gene reads) of methanotrophs (A-E) and methanogens (F-J) at different depths in the sediment at the five study stations shown in the order of increasing OM lability (from left to right; St3, St2, St4, St1, and St5) based on the surface sediment C:N ratio (see Table 1).

Methylomirabilis and ratio of *Ca*. Methylomirabilis to *Methylococcales* alongside the OM lability gradient (Table 2). Besides surface sediment C:N ratio, correlation analyses were conducted using porewater DIC concentrations, which also indicate sediment OM lability (i.e., increasing DIC indicate increasing OM lability), with results identical to those with C:N ratios (Table 2). Hence, in accordance with He et al. (25) and van Grinsven et al. (22), our results suggest that increasing input of labile OM leads to outcompetition of *Ca*. Methylomirabilis sp. by *Methylococcales* in lake sediments, which is very likely explained by their different metabolisms. While *Ca*. Methylomirabilis sp. bacteria are restricted in using $NO_2^-$ as an EA in $CH_4$ oxidation and have been reported to have a high affinity for $CH_4$ (10, 28), *Methylococcales* are potentially capable of coupling $CH_4$ oxidation with fermentation and with reduction of a variety of EAs (e.g., $O_2$, $NO_3^-$, $NO_2^-$, $Fe^{3+}$, and organic EAs) and have a low affinity for $CH_4$ (11–19, 26, 27). This gives *Methylococcales* advantage over *Ca*. Methylomirabilis in conditions of increased OM lability, when redox potential and availability of EAs is decreased and availability of $CH_4$ is increased.

## Methanogenic community

In accordance with lower redox potential and more suitable conditions for methanogenesis at stations with higher OM lability, the relative abundance of methanogens generally increased when the sediment C:N ratio decreased (Table 2; Fig. 2F through J; Fig. 3C). Methanogens were present all through the 0–10 cm layer at all stations. However, they had very low relative abundance in the surface 0–5 cm layer at all stations except for Station 5 with the highest sediment OM lability. This agrees with the $CH_4$ profile data described above, further suggesting that the methanogenesis zone extends closer to the sediment surface at stations with high OM lability (Fig. 1E and 2J). Our results agree with those of Yang et al. (23) showing a higher density of methanogens in sediments of an eutrophic than a mesotrophic lake. Based on the study by D'Ambrosio and Harrison (20) on the effect of eutrophication increasing the $CH_4$ emissions from lake sediments due to lowered $CH_4$ oxidation efficiency at increased methanogenesis rate, we hypothesized that increasing sediment OM lability would similarly lead to decreased abundance of methanotrophs in relation to methanogens. As above, we considered the top 0–3 cm $CH_4$ filter layer for methanotrophs in these analyses. For methanogens, we chose the layer 4–10 cm as it is below the $CH_4$ filter layer, and the relative abundance of methanogens is clearly highest below than above 4 cm depth at all stations except Station 5 (Fig. 2F through J). However, due to methanogens being abundant also above 4 cm depth at Station 5, we did the analyses for methanogens also by considering the whole 0–10 cm layer. In support of our hypothesis, the ratio of relative abundance of methanotrophs (sum of all methanotrophs) in the top 0–3 cm aerobic $CH_4$ filter layer to the relative abundance of methanogens in the 4–10 cm or 0–10 cm layer decreased when sediment C:N ratio decreased (Table 2). More specifically, this was due to the fact that the ratio *Ca*. Methylomirabilis to methanogens decreased strongly when the sediment C:N ratio decreased (Table 2; Fig. 3D). Similar to the hypothesis testing considering methanotrophs (see above), to increase robustness of the results, these analyses were carried out using both the average and the maximum relative abundance of methanotrophs and methanogens within the chosen layers (i.e., 0–3 cm for methanotrophs and 4–10 cm or 0–10 cm for methanogens), with similar results for the change in the ratio of *Ca*. Methylomirabilis to methanogens alongside the OM lability gradient (Table 2). As above for the hypothesis testing considering methanotrophs, the correlation analyses were also conducted using porewater DIC concentrations, with results identical to those with C:N ratios (Table 2). Hence, based on our results, it can be suggested that increasing input of labile OM changes lake sediment microbial community toward a lower genetic potential for methanotrophy relative to the genetic potential for methanogenesis (Table 2; Fig. 3). Further studies are required to assess whether the observed changes in the genetic potential have any role in controlling the sediment-to-water $CH_4$ emissions or whether the $CH_4$ emissions are solely determined by the activity of methanotrophs and methanogens.

**TABLE 2** Spearman correlation analysis results[a] on the relationship between microbial variables and environmental variables indicating sediment OM lability, i.e., C:N ratio[b] of the surface sediments (representing C:N ratio of sedimenting OM) and concentration of porewater dissolved inorganic carbon (DIC)[c]

| Microbial variables | C:N[b] (aver.) | C:N[b] (max.) | DIC[c] (aver.) | DIC[c] (max.) |
|---|---|---|---|---|
| **Relative abundances:** | | | | |
| Total methanotrophs (0–3 cm) | 0.7 | **0.9** | −0.7 | **−0.9** |
| *Methylococcales* (0–3 cm) | −0.7 | −0.6 | 0.7 | 0.6 |
| *Ca.* Methylomirabilis (0–3 cm) | **0.9** | **0.9** | **−0.9** | **−0.9** |
| *Ca.* Methanoperedens (0–3 cm) | 0.1 | −0.2 | −0.1 | 0.2 |
| Total methanogens (4–10 cm) | **−0.9** | −0.7 | **0.9** | 0.7 |
| Total methanogens (0–10 cm) | **−0.9** | −0.7 | **0.9** | 0.7 |
| **Ratios of relative abundances:** | | | | |
| *Ca.* Methylomirabilis (0–3 cm)/*Methylococcales* (0–3 cm) | 1 | 1 | −1 | −1 |
| Total methanotrophs (0–3 cm)/Total methanogens (4–10 cm) | **0.9** | 0.6 | **−0.9** | −0.6 |
| Total methanotrophs (0–3 cm)/Total methanogens (0–10 cm) | **0.9** | 0.6 | **−0.9** | −0.6 |
| *Methylococcales* (0–3 cm)/Total methanogens (4–10 cm) | 0.1 | −0.4 | −0.1 | 0.4 |
| *Methylococcales* (0–3 cm)/Total methanogens (0–10 cm) | 0.1 | −0.4 | −0.1 | 0.4 |
| *Ca.* Methylomirabilis (0–3 cm)/Total methanogens (4–10 cm) | 1 | 1 | −1 | −1 |
| *Ca.* Methylomirabilis (0–3 cm)/Total methanogens (0–10 cm) | 1 | 1 | −1 | −1 |

[a]$n=5$, statistically significant results ($P < 0.05$) highlighted in bold. Negative and positive correlations with C:N ratio and positive and negative correlation with DIC indicate increase and decrease, respectively, with increasing sediment OM lability. Aver. and max in brackets below the column title denote whether average or maximum relative abundance, respectively, in 0–3 cm (for methanotrophs), 4–10 cm or 0–10 cm sediment layers (for methanogens) was used in the analysis (see text).

[b]Using either C:N ratio in 0–1 cm sediment layer or C:N ratio of 0–2 cm sediment layer (i.e., average C:N ratio of 0–1 cm and 1–2 cm sediment layers) gave identical results. The C:N ratios are shown in Table 1.

[c]Average porewater DIC concentration within the 0–10 cm sediment layer. The values are shown in Table 1.

The order *Methanomassiliicoccales*, whose cultivated members use the methyl reducing methanogenic pathway (5), dominated the methanogen communities at stations with the lowest OM lability in sediment (Stations 3 and 2), while in addition to these, other methanogenic groups, such as the hydrogenotrophic ($H_2$ consuming) $CO_2$ reducing *Methanomicrobiales* (48), were relatively abundant members of methanogenic community at stations with higher OM lability (Stations 4, 1, and 5) (Fig. 2F through J). Also, acetoclastic (acetate using) *Methanosaetaceae* (48) was relatively abundant in Stations 4 and 5 (Fig. 2H and J).

The dominance of *Methanomassiliicoccales* is surprising because methanogenesis in freshwater sediments is typically dominated by the acetoclastic and hydrogenotrophic $CO_2$ reducing pathways (4). Yet, also in the eutrophic Lake Dianchi, *Methanomassiliicoccales* were dominant members of the methanogenic community, constituting at maximum over 20% of methanogens (23). However, recent metagenomic data suggested the presence of the acetoclastic pathway in genomes of uncultivated *Methanomassiliicoccales*, which could explain these findings (57). On the other hand, being only a very recently described order with limited information on its functional diversity (58), *Methanomassiliicoccales* might also contain non-methanogenic species. Therefore, the correlation analyses were made also by excluding *Methanomassiliicoccales* from the sum of total methanogens, but it did not affect the outcome of the study (Table 2; Table S2). The higher methanogenic functional diversity due to the increasing presence of hydrogenotrophic, $CO_2$ reducing *Methanomicrobiales,* and aceticlastic *Methanosaetaceae* at stations with higher sediment OM lability (i.e., Stations 4, 1, and 5) further reflects a lower redox potential and more suitable conditions for methanogenesis at these stations (Fig. 2F through J).

## Conclusion

Our results on the porewater profiles of lake sediments suggest modified redox zonation caused by changing sediment OM lability. This was because increased OM lability likely enhanced OM processing and consumption of electron acceptors. We also provide evidence that changes in lake sediment OM lability affect the structure of the $CH_4$ cycling

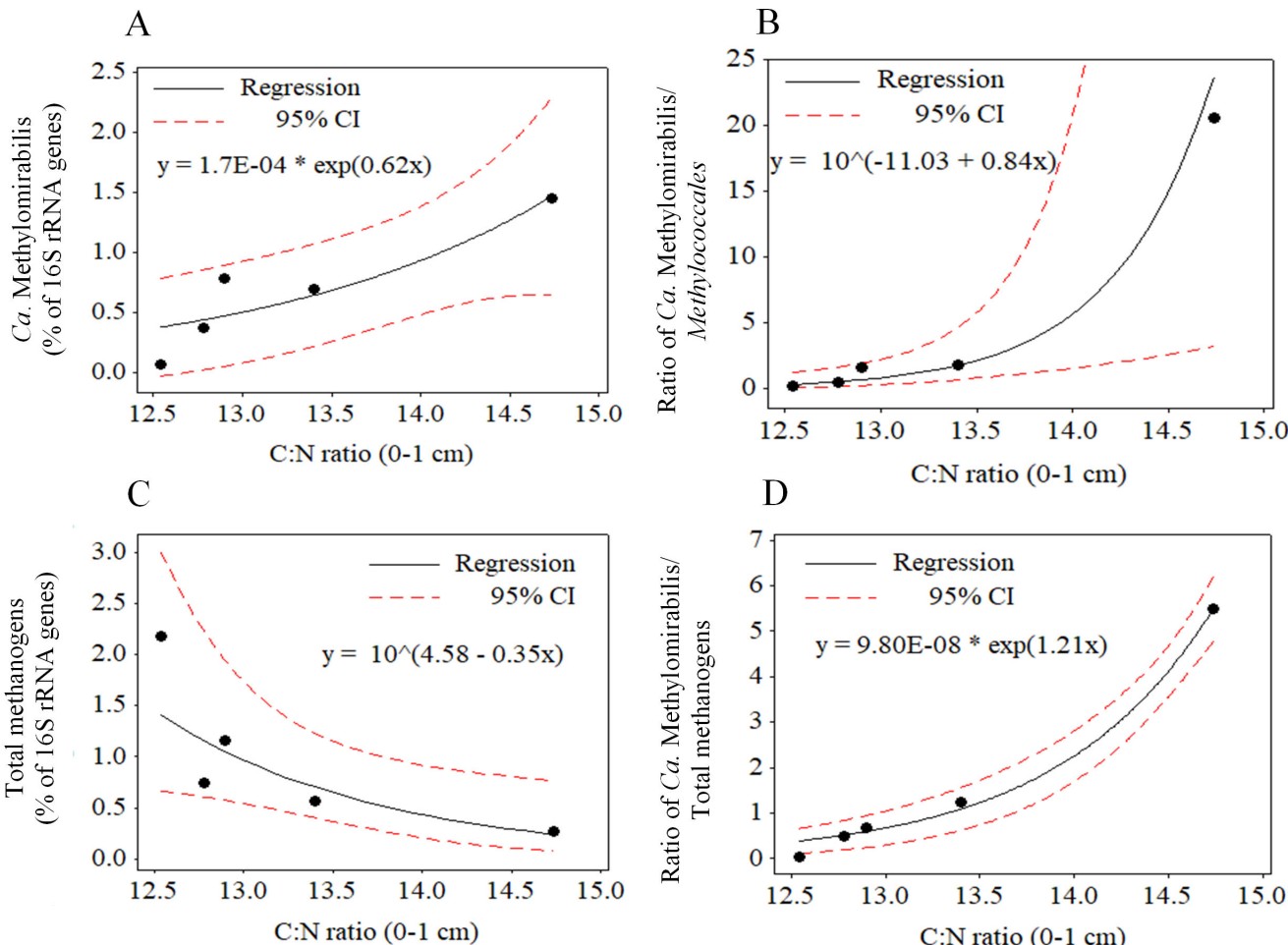

**FIG 3** Regression analysis results [observations in black dots, regression line in black line, and 95% confidence intervals (CIs) in red dot line] on the dependence of (A) the relative abundance of *Ca*. Methylomirabilis sp., (B) the ratio of abundances of *Ca*. Methylomirabilis to *Methylococcales*, (C) the relative abundance of total methanogens, and (D) the ratio of abundances of *Ca*. Methylomirabilis to total methanogens, with the surface sediment C:N ratios (i.e., C:N ratio of 0–1 cm layer). The average relative abundance of *Ca*. Methylomirabilis and *Methylococcales* within the 0–3 cm layer and the average relative abundance of methanogens within the 4–10 cm layer were used in the analyses. See Table S1 for detailed statistics of the regression models.

microbial community. By generating suitable conditions for methanogenesis (i.e., higher availability of methanogenic substrates and decreased availability of alternative EAs), increasing sediment OM lability increased the relative abundance of methanogens. Furthermore, increasing OM lability caused changes in the structure of the $CH_4$ filtering methanotrophic community by reducing the abundance of anaerobic nitrite-reducing methanotrophic *Ca*. Methylomirabilis sp. in relative to aerobic methanotrophic *Methylococcales* in lake sediments, which suggests that *Methylococcales* thrived better than *Ca*. Methylomirabilis under decreasing redox conditions and increasing methane availability due to their more diverse metabolism (fermentation and anaerobic respiration of various EAs) and lower affinity for $CH_4$. The results also suggest that increased OM lability decreases the abundance of methanotrophs, especially *Ca*. Methylomirabilis, relative to methanogens. Altogether, our results suggest that increasing input of labile algae-based OM changes the redox zonation of the sediment and exerts significant changes on the methanotrophic and methanogenic microbial community of lake sediments.

## ACKNOWLEDGMENTS

We thank the staff at the Lammi Biological Station for their support in field and laboratory work. Reviewers are acknowledged for their valuable comments and suggestions that improved the paper.

This study was supported by Academy of Finland (Grant No. 286642, 346751, and 353750 for A.J.R., 323214 for R.M., 307331 for H.J., and 310302 for S.L.A.), joint funding by Olvi Foundation, Jenny and Antti Wihuri Foundation, and Saastamoinen Foundation (for H.J.) and Kone Foundation (Grant No. 201803224 for A.J.R.), and a Tenure Track starting package from University of Helsinki (T.J.). Open access funding was provided by Tampere University.

Author contributions: Antti J. Rissanen: Conceptualization, Methodology, Formal analysis, Investigation, Funding acquisition, Project administration, Writing—original draft, Writing— review and editing; Tom Jilbert: Conceptualization, Methodology, Formal analysis, Investigation, Funding acquisition, Writing—review and editing; Asko Simojoki: Methodology, Formal analysis, Writing—review and editing; Rahul Mangayil: Methodology, Formal analysis, Writing—review and editing; Sanni L. Aalto: Methodology, Formal analysis, Writing—review and editing; Ramita Khanongnuch: Methodology, Formal analysis, Writing—review and editing Sari Peura: Methodology, Formal analysis, Writing—review and editing; Helena Jäntti: Conceptualization, Methodology, Formal analysis, Investigation, Funding acquisition, Project administration, Writing—review and editing.

## AUTHOR AFFILIATIONS

[1]Faculty of Engineering and Natural Sciences, Tampere University, Tampere, Finland
[2]Natural Resources Institute Finland (Luke), Helsinki, Finland
[3]Environmental Geochemistry Group, Department of Geosciences and Geography, Faculty of Science, Helsinki, Finland
[4]Department of Agricultural Sciences (Environmental Soil Science), Faculty of Agriculture and Forestry, University of Helsinki, Helsinki, Finland
[5]Department of Environmental and Biological Sciences, University of Eastern Finland, Kuopio, Finland
[6]Department of Biological and Environmental Sciences, University of Jyväskylä, Jyväskylä, Finland
[7]Department of Forest Mycology and Plant Pathology, Science for Life Laboratory, Swedish University of Agricultural Sciences, Uppsala, Sweden

## PRESENT ADDRESS

Rahul Mangayil, Department of Bioproducts and Biosystems, School of Chemical Engineering, Aalto University, Aalto, Finland
Sanni L. Aalto, Technical University of Denmark, DTU Aqua, Section for Aquaculture, The North Sea Research Centre, Hirtshals, Denmark

## AUTHOR ORCIDs

Antti J. Rissanen  http://orcid.org/0000-0002-5678-3361

## FUNDING

| Funder | Grant(s) | Author(s) |
| --- | --- | --- |
| Academy of Finland (AKA) | 286642, 346751, 353750 | Antti J. Rissanen |
| Academy of Finland (AKA) | 323214 | Rahul Mangayil |
| Academy of Finland (AKA) | 307331 | Helena Jäntti |
| Academy of Finland (AKA) | 310302 | Sanni L. Aalto |
| OLVI-Säätiö (Olvi Foundation) | | Helena Jäntti |

| Funder | Grant(s) | Author(s) |
|---|---|---|
| WRI \| Jenny ja Antti Wihurin Rahasto (Jenny JA Antti Wihurin Rahasto sr) | | Helena Jäntti |
| Saastamoisen säätiö (Saastamoinen Foundation) | | Helena Jäntti |
| Koneen Säätiö (Kone Foundation) | 201803224 | Antti J. Rissanen |
| University of Helsinki | | Tom Jilbert |

## DATA AVAILABILITY

The reads of the 16S rRNA gene amplicon data set are deposited in NCBI´s Sequence Read Archive (SRA) under the accession number PRJNA627335 for Stations 1–3 and PRJNA924099 for Stations 4 and 5. The other data are available in the supplemental data sets (see Supplemental Material).

## ADDITIONAL FILES

The following material is available online.

### Supplemental Material

**Supplemental methods, tables, and figures (Spectrum01955-23-S0001.doc).** Appendix A: Supplemental methods, Tables S1 and S2, Figures S1 and S2.
**Datasets (Spectrum01955-23-S0002.xlsx).** Contains all datasets except for 16S rRNA gene sequence data, which has been stored in NCBI´s SRA database.

### Open Peer Review

**PEER REVIEW HISTORY (review-history.pdf).** An accounting of the reviewer comments and feedback.

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
