## [Reviewer comments · Microbiology Spectrum]

Microbiology Spectrum

Organic matter lability modifies the vertical structure of methane-related microbial communities in lake sediments

Antti Rissanen, Tom Jilbert, Asko Simojoki, Rahul Mangayil, Sanni Aalto, Ramita Khanongnuch, Sari Peura, and Helena Jääntti

Corresponding Author(s): Antti Rissanen, Tampereen yliopisto - Hervannan kampus

Review Timeline:

Submission Date:	May 12, 2023
Editorial Decision:	June 8, 2023
Revision Received:	June 28, 2023
Accepted:	July 17, 2023

Editor: Sandi Orlic

Reviewer(s): Disclosure of reviewer identity is with reference to reviewer comments included in decision letter(s). The following individuals involved in review of your submission have agreed to reveal their identity: Lisa Y Stein (Reviewer #2)

Transaction Report:

DOI: <https://doi.org/10.1128/spectrum.01955-23>

June 8, 2023

Dr. Antti Juhani Rissanen
Tampereen yliopisto - Hervannan kampus
Faculty of Engineering and Natural Sciences
Korkeakoulunkatu 6
Tampere FI-33720
Finland

Re: Spectrum01955-23 (Organic matter lability modifies the vertical structure of methane-related microbial communities in lake sediments)

Dear Dr. Antti Juhani Rissanen:

Thank you for submitting your manuscript to Microbiology Spectrum. As you will see your paper is very close to acceptance. Please modify the manuscript along the lines I have recommended. As these revisions are quite minor, I expect that you should be able to turn in the revised paper in less than 30 days, if not sooner. If your manuscript was reviewed, you will find the reviewers' comments below.

When submitting the revised version of your paper, please provide (1) point-by-point responses to the issues raised by the reviewers as file type "Response to Reviewers," not in your cover letter, and (2) a PDF file that indicates the changes from the original submission (by highlighting or underlining the changes) as file type "Marked Up Manuscript - For Review Only". Please use this link to submit your revised manuscript. Detailed instructions on submitting your revised paper are below.

Link Not Available

Sincerely,

Sandi Orlic

Reviewer comments:

Reviewer #1 (Comments for the Author):

The authors have added necessary information to the methods that clarifies experimental design. However, I still contend that caveats need to be placed on the use of C:N ratios as metrics of OM lability. It is clear that there are patterns between C:N ratios and microbial groups, but the references used that make the case for using C:N ratios for OM lability and the influence of eutrophication have much larger differences (5-45), while the C:N ratios reported here range between 12.9 to 14.74. It is not clear that this is an ecologically meaningful range for OM lability or eutrophication, and that relationships are not simply sampling artifacts. The sample size is very low, and so statistical methods and interpretations must be robust. The authors indicate DIC measures are correlated with C:N. Is DIC also correlated with methanogen or methanotroph abundances? Clarifying these points would build a stronger case for the point that microbial groups vary with OM lability. If not, authors should only make the conclusions about C:N ratios, and not OM lability. Further, it is not clear that these differences in C:N ratios have implications towards changing eutrophication. I still contend that these conclusions are overstated.

Reviewer #2 (Comments for the Author):

This is a nicely done study showing how increased OM lability directly affects methane cycling microbial communities in lake sediments. There are a few conclusions that should be modified based on what has been reported in the literature regarding the function of methanotrophs.

line 87 - it is well documented in the literature that methanotrophs in the *Methylomonas* (e.g. *Methylomonas denitrificans*) and *Methylomicrobium* genera can use alternate terminal electron acceptors (e.g. nitrate, nitrite) as well substrates from fermentation of formaldehyde to survive and thrive in anoxic lakes and sediments. Thus, these MOB groups should not be considered obligately aerobic, but rather facultative. This fact should be mentioned here with proper citations (which you use further on) as it may explain the dynamics of *Methylomonas* spp. found in your data set. For instance, your second hypothesis might be valid only because MOB have more diverse metabolic pathways than *Ca. Methylomirabilis* and are thus much more competitive in high methane, anoxic, ecosystems. The way the hypotheses and data interpretation are presented in the manuscript does not feature the well documented metabolic capacity of *Methylomonas* spp. to denitrify and ferment in the absence of measurable oxygen.

line 206 -for the future, to be up to date, your amplicon analysis should be classifying sequences into ASVs and should be done using the Silva 138 database. OTUs and Silva 132 are outdated.

line 344 - 349 - the last sentence in this part of the paragraph should be the first sentence as it explains that *Methylococcales* use nitrate and fermentation in these ecosystems and not just oxygen and nitrite as EAs. This sentence and set of references should also be present in the introduction (as mentioned above) to better frame the study. The diversity of metabolism and low K_m for methane are likely better reasons for the higher competitiveness of *Methylococcales* over *Methylomirabilis*, and not just access to nitrite. The authors should also mention that *Methylococcales* can ferment formaldehyde and survive anoxia, which can further explain their competition for the same niche as *Methylomirabilis* spp.

line 377-380 - this statement about "genetic potential" being the driver for higher methanogenesis and lower methanotrophy is not supported, as kinetics of methane cycling is much more important than genetics. *Methylomirabilis* definitely does not support a higher methane consumption rate than *Methylococcales*; hence, the decrease in *Methylomirabilis* is inconsequential to the overall methane consumption rate. Rather, the increase in labile carbon, decrease in redox potential, increase in fermentation, and increase in available substrates for methanogenesis (particularly acetoclastic methanogenesis) is likely the bigger driver. It's also likely that rates of methane oxidation actually increased under the low C:N situation, but since methanogenesis rates increased more strongly, the result of methane efflux occurs. There is zero indication that any of this change in methane efflux is controlled by the ratio of *Methylomirabilis* and *Methylococcales*. This has much more to do with the increase in rates of methanogenesis and abundance of methanogens.

Please change the rationale/conclusion on line 413-415 as this observation is likely unrelated to nitrite competition and much more related to kinetics of methane oxidation (better for *Methylococcales*) and functional diversity of *Methylococcales* over *Methylomirabilis*. Also, the dominance of *Methylococcales* might have increased methane oxidation rates in some stations, but methanogenesis eclipsed this observation at stations with greatest rates of OM degradation.

Line 420 - this needs to change as genetic potential of methanotrophy resulting in differences in methane consumption rates is not validated in this or other studies; the rate of methane consumption is mainly kinetic and competitive at the enzyme level.

The data in Table 2 is confusing. Can the data be reorganized so that the max and average are reported together for a single measurement? Also, it would be easier to understand if the actual numerical ratios were reported instead of only correlation coefficients. The data from this table are not related to the corresponding information in the text, as the text refers to relative abundances in addition to whether the abundances correlate. So the table should also show relative abundances. The two columns for C:N 0-1cm and C:N 0-2cm don't make sense for the methanogen ratios, as these are sampled at 4-10cm. So how do the columns and rows match up for the methanogens when the sampling depths are different?

The data in Fig 2 show more support for methanogenesis overcoming methanotrophy due mostly to redox. For instance, the decrease in overall methanotroph gene abundances is accompanied by a massive increase in methanogen gene abundances at shallow depths. Thus, redox and increased substrates for methanogenesis is the major factor (presumably) controlling the rate in methane production versus consumption.

Data in Fig. 3 also indicate that *Methylomirabilis* is more competitive when methanogenesis and *Methylococcales* are low; which is when C:N is high and there is less OM degradation. Thus, it is incorrect to suggest that loss of *Methylomirabilis* means loss of methanotrophy; rather, the conclusion seems to be that *Methylomirabilis* is eclipsed by the more eutrophic-resistant *Methylococcales*. But again, *Methylococcales* have higher rates of methane consumption, so you can't conclude that less methanotrophy is occurring at low C:N, but rather that redox and higher acetate favor very high rates of methanogenesis regardless of the rate of methanotrophy.

Preparing Revision Guidelines

Please return the manuscript within 60 days; if you cannot complete the modification within this time period, please contact me. If you do not wish to modify the manuscript and prefer to submit it to another journal, please notify me of your decision immediately so that the manuscript may be formally withdrawn from consideration by Microbiology Spectrum.

Reviewer comments **and our answers:**

Reviewer #1 (Comments for the Author):

The authors have added necessary information to the methods that clarifies experimental design. However, I still contend that caveats need to be placed on the use of C:N ratios as metrics of OM lability. It is clear that there are patterns between C:N ratios and microbial groups, but the references used that make the case for using C:N ratios for OM lability and the influence of eutrophication have much larger differences (5-45), while the C:N ratios reported here range between 12.9 to 14.74. It is not clear that this is an ecologically meaningful range for OM lability or eutrophication, and that relationships are not simply sampling artifacts. The sample size is very low, and so statistical methods and interpretations must be robust. The authors indicate DIC measures are correlated with C:N. Is DIC also correlated with methanogen or methanotroph abundances? Clarifying these points would build a stronger case for the point that microbial groups vary with OM lability. If not, authors should only make the conclusions about C:N ratios, and not OM lability. Further, it is not clear that these differences in C:N ratios have implications towards changing eutrophication. I still contend that these conclusions are overstated.

ANSWER: Thank you for the comments. C:N ratio is “perfectly” negatively correlated with DIC concentration in Spearman rank correlation test ($\rho = -1.00$, $p < 0.0001$). This means that increasing DIC (coming from increased OM degradation) also tells about the increasing OM lability. Correlation between C:N ratio and DIC as well as the correlation between C:N ratio and acetate conc, and between C:N ratio and Shannon diversity index have now been added to the text to strengthen the conclusion that there is indeed variation in OM lability between the study stations . See lines 271-277 and 310-312.

As per referee suggestion, we have now also added the correlation analysis results on correlation between DIC and microbial variables into Table 2 (see new Table 2) and reported the results in lines 356-359 and 397-399. As seen in the new Table 2, The results between correlation analyses for C:N ratio vs. microbial variables and DIC vs. microbial variables are identical, further highlighting that it is indeed the variation in sediment OM lability affecting the changes in methane-cycling community.

However, referee has a very good point in saying whether the differences in sediment OM lability have implications towards changing eutrophication. Actually, we are not sure about that. Therefore, we have decided to tone down our conclusions regarding eutrophication. In the revised version, we conclude that changing sediment OM lability modifies sediment methane-cycling community, as this is supported by our data, but we do not anymore make conclusions on the effects of eutrophication. See for example abstract lines 56-58: “We conclude that increasing input of labile OM, subsequently affecting the redox zonation of sediments, significantly modifies the methane producing and consuming microbial community of lake sediments. “ and conclusions lines 442-444: “Altogether, our results suggest that increasing input of labile algae-based OM changes the redox zonation of the sediment and exert significant changes on the methanotrophic and methanogenic microbial community of lake sediments. “

Reviewer #2 (Comments for the Author):

This is a nicely done study showing how increased OM lability directly affects methane cycling microbial communities in lake sediments. There are a few conclusions that should be modified based

on what has been reported in the literature regarding the function of methanotrophs.

ANSWER: Thank you for the positive and constructive comments. We revised the manuscript according to the suggestions.

line 87 - it is well documented in the literature that methanotrophs in the *Methylomonas* (e.g. *Methylomonas denitrificans*) and *Methylomicrobium* genera can use alternate terminal electron acceptors (e.g. nitrate, nitrite) as well substrates from fermentation of formaldehyde to survive and thrive in anoxic lakes and sediments. Thus, these MOB groups should not be considered obligately aerobic, but rather facultative. This fact should be mentioned here with proper citations (which you use further on) as it may explain the dynamics of *Methylomonas* spp. found in your data set. For instance, your second hypothesis might be valid only because MOB have more diverse metabolic pathways than *Ca. Methylomirabilis* and are thus much more competitive in high methane, anoxic, ecosystems. The way the hypotheses and data interpretation are presented in the manuscript does not feature the well documented metabolic capacity of *Methylomonas* spp. to denitrify and ferment in the absence of measurable oxygen.

ANSWER: Thank you for bringing this up. Indeed, the more diverse metabolic pathways of MOB (especially *Methylococcales*) should be mentioned already in introduction. And as suggested, this is also very relevant considering our second hypothesis. We have revised the text all through according to this change in thinking.

See lines 48-53: "When OM lability increased, the abundance of anaerobic nitrite-reducing methanotrophs (*Candidatus Methylomirabilis*) relative to aerobic methanotrophs (*Methylococcales*) in the methane oxidation layer of sediment surface decreased, suggesting that *Methylococcales* were more competitive than *Ca. Methylomirabilis* under decreasing redox conditions and increasing methane availability due to their more diverse metabolism (fermentation and anaerobic respiration) and lower affinity for methane."

See lines 88-91: "Furthermore, MOB belonging to *Methylococcales* have been shown to be capable of metabolizing CH₄ in hypoxic and anoxic conditions via fermentation and anaerobic respiration of various EAs, including NO₃⁻, NO₂⁻, Fe³⁺, and organic EAs (11–19)."

See lines 111-116: "This is possibly due to MOB, especially *Methylococcales*, having a lower affinity for CH₄ and more diverse metabolism in hypoxic and anoxic conditions (incl. capability for fermentation and anaerobic respiration of various EAs as explained above) than *Ca. Methylomirabilis* (12, 15, 17, 18, 26–28). Hence, it could be expected that changes in the OM quality of lake sediments affect differently the abundances of MOB and *Ca. Methylomirabilis* sp. bacteria."

See lines 359-368: "Hence, in accordance with He et al. (25) and van Grinsven et al. (22), our results suggest that increasing input of labile OM leads to outcompetition of *Ca. Methylomirabilis* sp. by *Methylococcales* in lake sediments, which is very likely explained by their different metabolisms. While *Ca. Methylomirabilis* sp. bacteria are restricted in using NO₂⁻ as an EA in CH₄ oxidation and have been reported to have a high affinity for CH₄ (10, 28), *Methylococcales* are potentially capable of coupling CH₄ oxidation with fermentation and with reduction of a variety of EAs (e.g., O₂, NO₃⁻, NO₂⁻, Fe³⁺, and organic EAs) and have a low affinity for CH₄ (11–19, 26, 27). This gives *Methylococcales* advantage over *Ca. Methylomirabilis* in conditions of increased OM lability, when redox potential and availability of EAs is decreased and availability of CH₄ is increased."

See lines 434-440: "Furthermore, increasing OM lability causes changes in the structure of the CH₄ filtering methanotrophic community by reducing the abundance of anaerobic nitrite-reducing methanotrophic *Ca. Methylomirabilis* sp. in relative to aerobic methanotrophic *Methylococcales* in lake sediments, which suggests that *Methylococcales* thrived better than *Ca. Methylomirabilis*

under decreasing redox conditions and increasing methane availability due to their more diverse metabolism (fermentation and anaerobic respiration of various EAs) and lower affinity for CH₄.”

line 206 -for the future, to be up to date, your amplicon analysis should be classifying sequences into ASVs and should be done using the Silva 138 database. OTUs and Silva 132 are outdated.

ANSWER: Thank you for the comment. We agree about using Silva 132 database. In our current works, we already use the latest Silva database. However, as reported in supplementary methods, for this work, we did make a slight update to Silva 132. We supplemented it with a 16S rRNA gene sequence of *Candidatus Methylophilus sp.* (Methylococcales), which we have previously named and found to be abundant in boreal lakes.

For OTU vs ASV. We agree partially that ASV approach has advantages over OTU approach. Yet, in our case, we would not expect our results to differ between these two approaches, except in calculations of the Shannon diversity index (and the trend would be still very likely similar). This is because we use the relative abundances of certain taxonomic groups in our analyses. Although calculated from the OTU data, these relative abundances are actually determined by the number of 16S rRNA reads assigned to each taxonomic group.

line 344 - 349 - the last sentence in this part of the paragraph should be the first sentence as it explains that Methylococcales use nitrate and fermentation in these ecosystems and not just oxygen and nitrite as EAs. This sentence and set of references should also be present in the introduction (as mentioned above) to better frame the study. The diversity of metabolism and low K_m for methane are likely better reasons for the higher competitiveness of Methylococcales over *Methylomirabilis*, and not just access to nitrite. The authors should also mention that Methylococcales can ferment formaldehyde and survive anoxia, which can further explain their competition for the same niche as *Methylomirabilis* spp.

ANSWER: As per referees suggestion, we have now framed our study and hypothesis 2 by the metabolic diversity of Methylococcales vs. *Ca. Methylomirabilis*.

Furthermore, it is also likely that the different affinities for methane between *Ca. Methylomirabilis* and Methylococcales would also explain the differences in their distribution. This has now also been discussed. See above for answers to your first comment.

line 377-380 - this statement about "genetic potential" being the driver for higher methanogenesis and lower methanotrophy is not supported, as kinetics of methane cycling is much more important than genetics. *Methylomirabilis* definitely does not support a higher methane consumption rate than Methylococcales; hence, the decrease in *Methylomirabilis* is inconsequential to the overall methane consumption rate. Rather, the increase in labile carbon, decrease in redox potential, increase in fermentation, and increase in available substrates for methanogenesis (particularly acetoclastic methanogenesis) is likely the bigger driver. It's also likely that rates of methane oxidation actually increased under the low C:N situation, but since methanogenesis rates increased more strongly, the result of methane efflux occurs. There is zero indication that any of this change in methane efflux is controlled by the ratio of *Methylomirabilis* and Methylococcales. This has much more to do with the increase in rates of methanogenesis and abundance of methanogens.

ANSWER: Thanks for this notion. We have now removed the link between change in the genetic potential and estimated CH₄ fluxes. Indeed, kinetics of methane cycling could be much more important than genetics in explaining methane oxidation and methanogenesis and subsequent methane emission. Our data cannot tell if the genetic potential would have any controlling effect

on the CH₄ processes. We have modified this part according to referees suggestion. See lines 399-404: "Hence, based on our results, it can be suggested that increasing input of labile OM changes lake sediment microbial community towards a lower genetic potential for methanotrophy relative to the genetic potential for methanogenesis (Table 2, Fig. 3). Further studies are required to assess, whether the observed changes in the genetic potential have any role in controlling the sediment-to-water CH₄ emissions or whether the CH₄ emissions are solely determined by the activity of methanotrophs and methanogens."

Please change the rationale/conclusion on line 413-415 as this observation is likely unrelated to nitrite competition and much more related to kinetics of methane oxidation (better for Methylococcales) and functional diversity of Methylococcales over Methyloirabilis. Also, the dominance of Methylococcales might have increased methane oxidation rates in some stations, but methanogenesis eclipsed this observation at stations with greatest rates of OM degradation.

ANSWER: This has now been modified accordingly. See lines 434-440: "Furthermore, increasing OM lability causes changes in the structure of the CH₄ filtering methanotrophic community by reducing the abundance of anaerobic nitrite-reducing methanotrophic Ca. Methyloirabilis sp. in relative to aerobic methanotrophic Methylococcales in lake sediments, which suggests that Methylococcales thrived better than Ca. Methyloirabilis under decreasing redox conditions and increasing methane availability due to their more diverse metabolism (fermentation and anaerobic respiration of various EAs) and lower affinity for CH₄."

Line 420 - this needs to change as genetic potential of methanotrophy resulting in differences in methane consumption rates is not validated in this or other studies; the rate of methane consumption is mainly kinetic and competitive at the enzyme level.

ANSWER: We have revised this part accordingly. See lines 440-444: "The results also suggest that increased OM lability decreases the abundance of methanotrophs, especially Ca. Methyloirabilis, relative to methanogens. Altogether, our results suggest that increasing input of labile algae-based OM changes the redox zonation of the sediment and exert significant changes on the methanotrophic and methanogenic microbial community of lake sediments."

The data in Table 2 is confusing. Can the data be reorganized so that the max and average are reported together for a single measurement? Also, it would be easier to understand if the actual numerical ratios were reported instead of only correlation coefficients. The data from this table are not related to the corresponding information in the text, as the text refers to relative abundances in addition to whether the abundances correlate. So the table should also show relative abundances. The two columns for C:N 0-1cm and C:N 0-2cm don't make sense for the methanogen ratios, as these are sampled at 4-10cm. So how do the columns and rows match up for the methanogens when the sampling depths are different?

ANSWER: The data is now organized so that max and average are reported together in a single line. Unfortunately, we did not understand what was meant with the suggestion to report numerical ratios. This is a correlation table showing correlation between indicators of sediment OM lability, (i.e., C:N-ratio of surface sediment and in the revised table we also include the DIC concentration) and relative abundance of methane-cycling groups and abundance ratios (e.g. ratio of Ca. Methyloirabilis/Methylococcales) calculated using the relative abundances of the methane cycling groups. Hence, only the correlation coefficients are shown and as denoted, the significant correlations have been highlighted in bold. For C:N, we used values from the 0-1 cm layer as well as average of 0-1cm and 0-2cm layer due to reasons explained in text in lines 260-

263: "Surface sediment values were used here due to assumed minimum overprinting of sediment diagenetic processes on the C:N ratio, hence the values should represent the C:N ratio (lability) of the OM sedimenting to the lake bottom at the stations."

Hence, these surface sediment values were used, because they would represent the OM sedimenting to bottom, which is still not affected by mineralization processes. Hence, the surface sediment values are indicators of the lability of OM sedimenting to lake bottom. Because correlation analysis results were identical when using C:N of 0-1cm or C:N of 0-2 cm, in the revised table, we decided to report only one set of correlation coefficients for surface sediment C:N. See new table 2.

The data in Fig 2 show more support for methanogenesis overcoming methanotrophy due mostly to redox. For instance, the decrease in overall methanotroph gene abundances is accompanied by a massive increase in methanogen gene abundances at shallow depths. Thus, redox and increased substrates for methanogenesis is the major factor (presumably) controlling the rate in methane production versus consumption.

ANSWER: Thank you for this interpretation. As noted above, we have revised the paper.

Data in Fig. 3 also indicate that *Methylomirabilis* is more competitive when methanogenesis and *Methylococcales* are low; which is when C:N is high and there is less OM degradation. Thus, it is incorrect to suggest that loss of *Methylomirabilis* means loss of methanotrophy; rather, the conclusion seems to be that *Methylomirabilis* is eclipsed by the more eutrophic-resistant *Methylococcales*. But again, *Methylococcales* have higher rates of methane consumption, so you can't conclude that less methanotrophy is occurring at low C:N, but rather that redox and higher acetate favor very high rates of methanogenesis regardless of the rate of methanotrophy.

ANSWER: Thank you for this interpretation. As noted above, we have now revised the paper.

July 17, 2023

Dr. Antti Juhani Rissanen
Tampereen yliopisto - Hervannan kampus
Faculty of Engineering and Natural Sciences
Korkeakoulunkatu 6
Tampere FI-33720
Finland

Re: Spectrum01955-23R1 (Organic matter lability modifies the vertical structure of methane-related microbial communities in lake sediments)

Dear Dr. Antti Juhani Rissanen:

Your manuscript has been accepted, and I am forwarding it to the ASM Journals Department for publication. You will be notified when your proofs are ready to be viewed.

Sincerely,

Sandi Orlic
Editor, Microbiology Spectrum
